# The Effect of Annealing on the Soft Magnetic Properties and Microstructure of Fe_82_Si_2_B_13_P_1_C_3_ Amorphous Iron Cores

**DOI:** 10.3390/ma16165527

**Published:** 2023-08-09

**Authors:** Wei Zheng, Guangqiang Zhang, Qian Zhang, Haichen Yu, Zongzhen Li, Mingyu Gu, Su Song, Shaoxiong Zhou, Xuanhui Qu

**Affiliations:** 1Institute for Advanced Materials and Technology, University of Science and Technology Beijing, Beijing 100083, China; hy.zw@163.com (W.Z.); atzhangqian@163.com (Q.Z.); 2Central Iron and Steel Research Institute, Beijing 100081, China; 3Jiangsu JITRI Advanced Energy Materials Research Institute Co., Ltd., Changzhou 213001, China; 18601017215@126.com (G.Z.); zongzhenli@hotmail.com (Z.L.); songs@aemcn.com (S.S.); 4Institute of Advanced Materials, North China Electric Power University, Beijing 102206, China; yuhaichen@atmcn.com; 5Electrical & Information Engineering, Shandong University, Weihai 264209, China; gumingyu19980322@163.com

**Keywords:** amorphous iron core, magnetic field annealing, core loss, domains

## Abstract

This research paper investigated the impact of normal annealing (NA) and magnetic field annealing (FA) on the soft magnetic properties and microstructure of Fe_82_Si_2_B_13_P_1_C_3_ amorphous alloy iron cores. The annealing process involved various methods of magnetic field application: transverse magnetic field annealing (TFA), longitudinal magnetic field annealing (LFA), transverse magnetic field annealing followed by longitudinal magnetic field annealing (TLFA) and longitudinal magnetic field annealing followed by transverse magnetic field annealing (LTFA). The annealed samples were subjected to testing and analysis using techniques such as differential scanning calorimetry (DSC), transmission electron microscopy (TEM), X-ray diffraction (XRD), magnetic performance testing equipment and magneto-optical Kerr microscopy. The obtained results were then compared with those of commercially produced Fe_80_Si_9_B_11_. Fe_82_Si_2_B_13_P_1_C_3_ demonstrated the lowest loss of P_1.4T,2kHz_ = 8.1 W/kg when annealed in a transverse magnetic field at 370 °C, which was 17% lower than that of Fe_80_Si_9_B_11_. When influenced by the longitudinal magnetic field, the magnetization curve tended to become more rectangular, and the coercivity (B_3500A/m_) of Fe_82_Si_2_B_13_P_1_C_3_ reached 1.6 T, which was 0.05 T higher than that of Fe_80_Si_9_B_11_. During the 370 °C annealing process of the Fe_82_Si_2_B_13_P_1_C_3_ amorphous iron core, the internal stress in the strip gradually dissipated, and impurity domains such as fingerprint domains disappeared and aligned with the length direction of the strip. Consequently, wide strip domains with low resistance and easy magnetization were formed, thereby reducing the overall loss of the amorphous iron core.

## 1. Introduction

Soft magnetic materials and their related devices (inductors, transformers and electrical machines) play a key role in the conversion of energy throughout our world. The conversion of electrical power includes the bidirectional flow of energy between sources, storage and the electrical grid and is accomplished via the use of power electronics. Electrical machines (motors and generators) transform mechanical energy into electrical energy and vice versa. The introduction of wide bandgap (WBG) semiconductors is allowing power conversion electronics and motor controllers to operate at much higher frequencies. This reduces the size requirements for passive components (inductors and capacitors) in power electronics and enables more efficient, high-rotational-speed electrical machines [1]. Fe-based amorphous alloys exhibit superior soft magnetic properties, including relatively high saturated magnetization, high effective permeability, low coercivity and low core loss, especially at middle and high frequencies, compared to other traditional soft magnetic materials, such as ferrite, silicon steel and permalloy. These kinds of alloys with unique structures, excellent magnetic properties and good mechanical properties have attracted much attention in a variety of emerging science and engineering fields [2].

Amorphous alloy ribbons possess desirable properties, such as low losses and high magnetic permeability, making them potential replacements for non-oriented silicon steel in high-frequency motors. This substitution can reduce iron core losses by more than 80% and significantly enhance overall efficiency [3,4,5,6]. However, the saturation magnetic induction strength (B_s_) of commercially available amorphous ribbons is only 1.56 T, which is considerably lower than that of non-oriented silicon steel (1.8–2.0 T). This limitation hampers the application of amorphous ribbons in motors. Consequently, there is a pressing need to explore the composition design and applications of high-B_s_ amorphous alloys, which has become a prominent research and development focus in this field. Although there have been several relevant reports [7,8,9], most of them have been conducted in laboratory settings with a limited number of high-quality samples, which are often narrow in width. These samples are suitable for studying strip characteristics such as saturation magnetic induction and coercivity. However, they are inadequate for producing iron core samples required for the comprehensive testing and analysis of losses and magnetization curves.

Relaxation upon isothermal annealing below the crystallization transition temperature is applied to release the residual stress in metallic amorphous alloys [10] to enhance their soft magnetic performance [8]. Ordinarily, the soft magnetic properties of amorphous alloys can be further optimized through appropriate annealing characteristics, such as the annealing temperature, heating rate and holding time [11]. As-spun amorphous ribbons are usually subjected to magnetic field annealing to improve their soft magnetic properties (SMPs) [12]. Magnetic field annealing treatment can bring about a process of atomic pair ordering and the decline of the coercive field [13]. Magnetic field annealing has a positive effect on changing the domain-sensitive direction and unifying the easy magnetization direction [14]. Magnetic field annealing causes the formation of strip domains in the sample for optimal SMPs [15]. Magnetic field annealing can promote inner stress release and modulate the domain structure, which directly affects soft magnetic alloys [16]. However, the mechanism of the effect of magnetic field annealing on SMPs is not yet clear.

This paper focuses on the utilization of planar flow continuous casting technology to produce wide amorphous alloy ribbons and high-quality annular amorphous iron core samples based on the previously developed Fe_82_Si_2_B_13_P_1_C_3_ alloy, which possesses a high saturation magnetic induction strength. This study systematically investigates the impact of heat treatment processes on the soft magnetic properties of these amorphous iron cores, aiming to uncover the underlying mechanisms of magnetic domain morphology on the magnetic properties. By employing this approach, a heat treatment technique is developed for amorphous iron cores with a high saturation magnetic induction strength and low loss. Furthermore, systematic magnetization curves and loss data are obtained, which are expected to offer valuable insight for the application of Fe_82_Si_2_B_13_P_1_C_3_ amorphous ribbons in motors.

## 2. Methods

Sample source: The GFA and soft magnetic properties of as-spun Fe_80_Si_9_B_(11−x)_P_x_ amorphous alloys were improved due to the incorporation of a minor amount of P element. However, excess P led to a reduction in iron content and the deterioration of saturation magnetization [17]. Wide amorphous alloy ribbons with compositions of Fe_80_Si_9_B_11_ (1K101) and Fe_82_Si_2_B_13_P_1_C_3_ were produced using planar flow continuous casting. The ribbons had a width of 150 ± 0.5 mm and a thickness of 25 ± 1 µm. These wide ribbons were subsequently cut into 10 mm narrow ribbons and wound into circular amorphous samples. The resulting samples had an outer diameter of φ_outer_ = 35 mm, an inner diameter of φ_inner_ = 25 mm and a height of h = 10 mm.

Annealing experiments:

Normal annealing (NA): The amorphous iron core was positioned within the annealing area of the furnace. The furnace door was closed, and the interior was evacuated before being filled with nitrogen gas. The temperature of the furnace was then raised from room temperature to the designated holding temperature (Fe_80_Si_9_B_11_:370–420 °C and Fe_82_Si_2_B_13_P_1_C_3_:340–390 °C) at a heating rate of 6 °C/min. The temperature was maintained at the holding temperature for a duration of 40 min, followed by furnace cooling until reaching room temperature to allow for sample removal.

Magnetic field annealing (FA):

In the case of transverse magnetic field annealing (TFA), as depicted in Figure 1a, pure iron magnetic poles of equal inner and outer diameters and a length of 150 mm were affixed on both sides of the sample. Subsequently, the furnace door was closed, and the furnace was evacuated and filled with nitrogen. The heating rate was set at 6 °C/min. Upon reaching the desired temperature, a transverse magnetic field was applied with the specified current. After a holding period of 40 min, the sample was cooled to room temperature and removed from the furnace.

During longitudinal magnetic field annealing (LFA), as illustrated in Figure 1b, the sample was threaded to the center of the copper rod. The remaining procedures were identical to those of TFA, with the exception that a longitudinal magnetic field was applied.

Test method: The amorphous nature of the alloy strip was assessed using X-ray diffraction (XRD) with Cu Kα (λ = 0.15406 nm) radiation in the range of 30–90° taking 0.2 s for one step of 0.02°. FEI Strata 400S focused ion beam scanning electron microscopy (FIB-SEM) was used to cut the ribbon, and FEI Talos F200s transmission electron microscopy (TEM) was used to observe the cross-section of the ribbon (FEI, Los Angeles, CA, USA). Thermal parameters, including the crystallization temperature (T_x_) and crystallization peak (T_p_), were analyzed via differential thermal analysis (DSC) at a heating rate of 10 °C/min under the protection of argon gas to establish an appropriate range of annealing temperatures. The loss and hysteresis loop of the amorphous iron core were measured using both direct current (model TD8150) and alternating current (model TK7500) magnetic performance testing instruments. Additionally, a magneto-optical Kerr effect (MOKE, Evico 4-873K/950 MT) microscope was employed to examine the magnetic domain structures (evico magnetics GmbH, Dresden, Germany).

## 3. Results and Discussion

In this study, the magnetic properties of the novel amorphous strip (Fe_82_Si_2_B_13_P_1_C_3_) were systematically investigated under various heat treatment processes. The research aimed to understand the influence of magnetic fields on the morphology and orientation of magnetic domains and elucidate the underlying mechanisms.

### 3.1. Structure and Performance Analysis of Fe_82_Si_2_B_13_P_1_C_3_ Ribbons

The XRD pattern of the as-spun Fe_82_Si_2_B_13_P_1_C_3_ strip is presented in Figure 2a. The diffraction curve reveals a distinct diffuse peak near the diffraction angle of 2θ = 45°, and no sharp crystallization peaks are observed. This pattern confirms that the strip possesses an amorphous structure. Figure 2b displays the high-resolution TEM (HRTEM) image and the selected area electron diffraction (SAED) pattern of the as-spun Fe_82_Si_2_B_13_P_1_C_3_ strip. The atomic arrangement within the structure exhibits disorder, which is indicative of an amorphous structure. The presence of diffraction halos in the pattern further supports the amorphous nature of the strip.

Figure 2c presents the results of the DSC tests conducted on the Fe_82_Si_2_B_13_P_1_C_3_ and Fe_80_Si_9_B_11_ strip samples. Both samples exhibit two distinct exothermic peaks. The crystallization onset temperature (T_x1_) was determined by analyzing the tangent to the initial arc of the first exothermic peak. As indicated in Table 1, the T_x1_ of the Fe_82_Si_2_B_13_P_1_C_3_ strip was 28 °C lower than that of Fe_80_Si_9_B_11_. This observation suggests that the addition of P and C elements promotes the early precipitation of the α-Fe phase. The temperature range from the Curie temperature (T_c_) to T_x1_ is commonly considered an optimal heat treatment interval for amorphous ribbons. Furthermore, the ideal annealing temperature is typically achieved by reducing the temperature by 80–120 °C from T_x1_. These criteria serve as essential guidelines for determining the appropriate annealing temperature [18]. Figure 2c shows two crystallization peaks, with the first corresponding to the precipitation of the α-Fe phase and the second corresponding to the formation of the Fe–B phase [19,20]. Table 1 reveals that the crystallization peak temperatures of the Fe_82_Si_2_B_13_P_1_C_3_ strip are lower than those of Fe_80_Si_9_B_11_, and the gaps ΔT_p_ between the crystallization peaks remain relatively consistent for both ribbons.

The magnetization curve of the as-spun Fe_82_Si_2_B_13_P_1_C_3_ strip is depicted in Figure 2d. Under an excitation of 3500 A/m, the values of magnetic induction (B_m_) for Fe_82_Si_2_B_13_P_1_C_3_ and Fe_80_Si_9_B_11_ were measured to be 1.34 T and 1.31 T, respectively. The higher Fe content in Fe_82_Si_2_B_13_P_1_C_3_ accounts for its elevated B_m_ value.

### 3.2. Effects of NA Processes on Magnetic Properties of Amorphous Iron Cores

Based on the DSC test results, Fe_80_Si_9_B_11_ underwent annealing within the temperature range of 380–420 °C, and Fe_82_Si_2_B_13_P_1_C_3_ was annealed at temperatures between 350 and 390 °C. The XRD spectra of the amorphous iron cores are displayed in Figure 3a,b. From the figures, it can be observed that, when Fe_80_Si_9_B_11_ was annealed at ≤400 °C and Fe_82_Si_2_B_13_P_1_C_3_ was annealed at ≤370 °C, a diffuse diffraction peak appeared near 2θ = 45° for both alloys, indicating the retention of their amorphous state. However, when Fe_80_Si_9_B_11_ was annealed at ≥410 °C and Fe_82_Si_2_B_13_P_1_C_3_ was annealed at ≥380 °C, the diffuse peak became sharper, suggesting the precipitation of trace crystalline phases. These crystalline phases can be identified as α-Fe(Si) with a body-centered cubic (bcc) structure. Furthermore, the XRD spectra revealed that, with the increasing annealing temperature, the intensity of the diffraction peak corresponding to the α-Fe(Si) phase gradually strengthened, indicating an augmentation in the crystallization fraction of the α-Fe(Si) phase as the annealing temperature rose.

Figure 4a,b illustrate the loss curves of Fe_80_Si_9_B_11_ annealed at 370–410 °C and Fe_82_Si_2_B_13_P_1_C_3_ annealed at 340–380 °C at different frequencies. The curves display an initial decrease followed by an increase as the heat treatment temperature rises. According to Table 2, the minimum loss for Fe_80_Si_9_B_11_ was achieved at 400 °C, with a value of P_1.4T,2kHz_ = 36 W/kg. Similarly, Fe_82_Si_2_B_13_P_1_C_3_ reached its minimum loss at 370 °C, with a value of P_1.4T,2kHz_ = 29 W/kg. Additionally, as the excitation frequency increased, the losses also increased, with the loss at 2 kHz being nearly 100 times higher compared to that at 50 Hz. The losses for Fe_80_Si_9_B_11_ annealed at 410 °C and Fe_82_Si_2_B_13_P_1_C_3_ annealed at 380 °C were P_1.4T,2kHz_ = 58 W/kg and P_1.4T,2kHz_ = 47 W/kg, respectively. The XRD curves in Figure 3a,b exhibited crystallization peaks for Fe_80_Si_9_B_11_ ≥ 410 °C and Fe_82_Si_2_B_13_P_1_C_3_ ≥ 380 °C, providing further evidence of increased loss after the crystallization of the amorphous iron cores.

Figure 4c depicts the frequency-dependent loss curves of Fe_80_Si_9_B_11_ annealed at 400 °C and Fe_82_Si_2_B_13_P_1_C_3_ annealed at 370 °C under an excitation of 1.4 T. It is evident that, within the frequency range of 50 Hz to 2 kHz, Fe_82_Si_2_B_13_P_1_C_3_ exhibits lower losses compared to those of Fe_80_Si_9_B_11_ at 1.4 T.

Figure 4d displays the magnetization curves of Fe_80_Si_9_B_11_ annealed at 400 °C and Fe_82_Si_2_B_13_P_1_C_3_ annealed at 370 °C. Prior to the intersection point of the two curves (indicated by the dashed line) at an excitation of 130 A/m, the working magnetic flux density B_m_ of Fe_80_Si_9_B_11_ exceeded that of Fe_82_Si_2_B_13_P_1_C_3_. However, beyond the excitation of 130 A/m, the B_m_ of Fe_82_Si_2_B_13_P_1_C_3_ surpassed that of Fe_80_Si_9_B_11_. At an excitation of 3500 A/m, Fe_82_Si_2_B_13_P_1_C_3_ exhibited a B_m_ of 1.59 T, which was higher than that of Fe_80_Si_9_B_11_ (B_m_ = 1.55 T). The higher Fe content in Fe_82_Si_2_B_13_P_1_C_3_, as indicated in Table 2, contributed to its elevated B_m_ value beyond an excitation of 130 A/m.

In short, the Fe_82_Si_2_B_13_P_1_C_3_ amorphous iron core demonstrated a lower loss than that of Fe_80_Si_9_B_11_. The Fe_82_Si_2_B_13_P_1_C_3_ amorphous iron core in the motor has the advantage of high efficiency, high power density and high torque density. The Fe_82_Si_2_B_13_P_1_C_3_ amorphous iron core exhibited a higher B_m_ than that of Fe_80_Si_9_B_11_. Compared with Fe_80_Si_9_B_11_, the amorphous iron core possess a relatively lower B_m_, which hampers the miniature of the devices.

Figure 5a,b depict the magnetic domain morphologies of the as-spun Fe_80_Si_9_B_11_ and Fe_82_Si_2_B_13_P_1_C_3_ amorphous ribbons, respectively. Magnetic domains exhibit various types, such as sheet domains, closed domains, checkerboard domains, ripple domains, magnetic bubble domains and their derivatives [21]. Because the amorphous alloys lack grains, the influence of magnetic crystal anisotropy can be disregarded. Thus, the magnetic domain structure is primarily determined according to the residual stress generated during the preparation of the amorphous ribbons [22]. The high stress levels during the extremely cold process of quenching result in magnetic domains with irregular shapes, varying widths, disordered orientations and the presence of fine fingerprint domains. Residual stress has a detrimental effect on magnetic properties, and annealing has been proven to be effective in relieving internal stress and enhancing the magnetic properties of the ribbons [23]. Annealing induces structural relaxation and the release of internal stress in the amorphous strip, causing the magnetic domains within the strip to align along the direction of the easily magnetized axis, which is parallel to the strip’s length. This minimizes magnetic anisotropy within the strip [24]. The magnetic domain morphologies of Fe_80_Si_9_B_11_ annealed at 370 °C and Fe_82_Si_2_B_13_P_1_C_3_ annealed at 340 °C are illustrated in Figure 5c,d, respectively. The chosen annealing temperatures alleviate some stress, resulting in magnetic domains with converging orientations along the length direction of the ribbons. However, the shapes of the domains remain irregular and exhibit varying sizes. The results for Fe_80_Si_9_B_11_ annealed at 400 °C and Fe_82_Si_2_B_13_P_1_C_3_ annealed at 370 °C are presented in Figure 5e,f, respectively. Clearly, as the annealing temperature increases, wide and uniform magnetic domains form with consistent orientations within the ribbons. Moreover, Figure 5g,h demonstrate that, when Fe_80_Si_9_B_11_ was annealed at 410 °C and Fe_82_Si_2_B_13_P_1_C_3_ was annealed at 380 °C, the previously wide magnetic domains started to crack, becoming irregular and narrower. These observations indicate that the annealed amorphous ribbons began to crystallize at these temperatures.

Magnetic domain observations not only provide insight into the spatial distribution of domains within ferromagnetic materials but also serve as a foundation for investigating the magnetic properties of amorphous iron cores. During the magnetization process of these cores, the magnetic domain walls undergo bending, which can be impeded by impurities or internal stress within the material. To overcome these obstacles, partial excitation is employed to induce the bending of magnetic domain walls, resulting in their conversion into heat and subsequent energy losses. This motion of the domain walls generates eddy current losses. As the heat treatment temperature increases, the internal stress within the amorphous iron cores gradually diminishes, and impurities such as fingerprint domains gradually disappear. Consequently, the magnetic domains become more uniform, as illustrated in Figure 5e,f. In this scenario, wide strip domains characterized by low resistance and uniform changes are formed, leading to a reduction in the overall losses of the amorphous iron cores, as demonstrated by the red curves in Figure 4a,b.

### 3.3. Effects of FA on Magnetic Properties of Amorphous Iron Cores

Figure 6a illustrates the effect of the annealing temperature on the loss under a transverse magnetic field of 10.6 kA/m. The results indicate that the loss first decreases and then increases with the increasing annealing temperature. Specifically, the minimum loss for Fe_80_Si_9_B_11_, recorded at P_1.4T,2kHz_ = 12.3 W/kg, is achieved at an annealing temperature of 400 °C. Similarly, the lowest loss for Fe_82_Si_2_B_13_P_1_C_3_, measured at P_1.4T,2kHz_ = 11.3 W/kg, occurs at an annealing temperature of 370 °C. As the annealing temperature is further increased, the losses begin to rise, reaching P_1.4T,2kHz_ = 20.2 W/kg for Fe_80_Si_9_B_11_ annealed at 410 °C and P_1.4T,2kHz_ = 12.8 W/kg for Fe_82_Si_2_B_13_P_1_C_3_ annealed at 380 °C. Notably, the loss of Fe_82_Si_2_B_13_P_1_C_3_ in the case of TFA is significantly lower than that of Fe_80_Si_9_B_11_ (As shown by the dashed line). To further investigate the impact of changing the transverse magnetic field on the magnetic properties, different magnetic field intensities were applied to the Fe_82_Si_2_B_13_P_1_C_3_ amorphous iron core at an annealing temperature of 370 °C. Figure 6b displays the variations in the loss. With an increased magnetic field intensity of 31.8 kA/m, the loss further decreased to P_1.4T,2kHz_ = 8.1 W/kg, which is notably superior to the loss of the NA sample at 370 °C (P_1.4T,2kHz_ = 29 W/kg), as shown in Figure 4d.

This study also examined the impact of different annealing processes, including NA, TFA, LFA, TLFA and LTFA, on the magnetic properties of Fe_82_Si_2_B_13_P_1_C_3_ when annealed at 370 °C. Figure 6c demonstrates that the sample treated with TFA exhibited the lowest loss, measured at P_1.4T,2kHz_ = 8.1 W/kg, which was significantly superior to the losses observed in the other annealing processes. Conversely, LFA showed a poor loss performance. LTFA resulted in a loss of P_1.4T,2kHz_ = 11.9 W/kg, which was better than that of TLFA (P_1.4T,2kHz_ = 30.7 W/kg), as outlined in Table 3.

Figure 7a,b present the magnetic domain morphologies of Fe_82_Si_2_B_13_P_1_C_3_ annealed at 370 °C under transverse and longitudinal magnetic fields, respectively. In the case of TFA, the magnetic domain morphology appeared finer and exhibited a greater consistency in direction, as depicted in Figure 7c,d. When the longitudinal magnetic field was applied first or later, an increase in Z-shaped bending within the magnetic domains was observed, with the magnetic domain morphology of LTFA appearing finer compared to that of TLFA.

In conclusion, the morphology and orientation of magnetic domains were found to be closely related to the magnetic properties of the material. Furthermore, the results suggest that the presence and alteration of elements have an impact on the morphology of magnetic domains, consequently influencing the magnetic properties of the material.

Figure 8 illustrates the magnetization curves of Fe_82_Si_2_B_13_P_1_C_3_ amorphous iron cores subjected to various treatments: NA, TFA, LFA, TLFA and LTFA at 370 °C. In Figure 8a, under an applied excitation of 3500 A/m, the B_m_ values of the treatment processes were arranged in ascending order as TFA < LTFA < NA < LFA < TLFA. Particularly, the B_m_ values of LFA and TLFA closely overlapped, both reaching 1.6 T. Figure 8b presents enlarged images at the inflection points. The ordering of B_m_ values under an excitation of 200 A/m was consistent with that under 3500 A/m. Notably, the LFA magnetization curve exhibited a distinct rectangular shape, enhancing the maximum permeability and residual magnetic induction strength while also increasing the loss. Conversely, TFA resulted in a flattened magnetization curve, promoting a more stable magnetic permeability, lower residual magnetic induction strength and reduced loss. The low-loss amorphous alloy iron core proves to be an exceptional material for pulse transformers. Magnetic field heat treatment can introduce uniaxial magnetic anisotropy to the amorphous iron core, enabling precise control of its size, direction and modulation of the magnetization curve to fulfill specific application requirements.

## 4. Conclusions

This study conducted a comprehensive investigation into the impact of NA and FA on the magnetic properties of an Fe_82_Si_2_B_13_P_1_C_3_ amorphous iron core and compared it with an Fe_80_Si_9_B_11_ amorphous iron core. The findings can be summarized as follows.

(1)The Fe_82_Si_2_B_13_P_1_C_3_ amorphous iron core, subjected to TFA, demonstrated a lower loss of P_1.4T,2kHz_ = 8.1 W/kg, which was 17% less than that of Fe_80_Si_9_B_11_ with a loss of P_1.4T,2kHz_ = 9.8 W/kg. Furthermore, the Fe_82_Si_2_B_13_P_1_C_3_ amorphous iron core exhibited a higher B_3500A/m_ value of 1.6 T under LFA, which was 0.05 T greater than the B_3500A/m_ value of Fe_80_Si_9_B_11_ (B_3500A/m_ = 1.55 T). The optimal heat treatment temperature for Fe_82_Si_2_B_13_P_1_C_3_ was 30 °C lower than that of Fe_80_Si_9_B_11_, contributing to improved toughness of the amorphous ribbons.(2)As the heat treatment temperature of Fe_82_Si_2_B_13_P_1_C_3_ increased to 370 °C, the internal stress gradually dissipated, resulting in the disappearance of impurity domains, such as fingerprint domains. Moreover, the magnetic domains within the strip underwent a transformation, aligning themselves with the length direction of the strip. This led to the formation of wide strip domains with low resistance, facilitating easy magnetization and ultimately reducing the overall loss of the amorphous iron core.(3)For Fe_82_Si_2_B_13_P_1_C_3_ amorphous iron cores under an applied excitation of 3500 A/m, the B_m_ values of the treatment processes were arranged in ascending order as TFA < LTFA < NA < LFA < TLFA. Particularly, the B_m_ values of LFA and TLFA closely overlapped, with both reaching 1.6 T.

## Figures and Tables

**Figure 1 materials-16-05527-f001:**
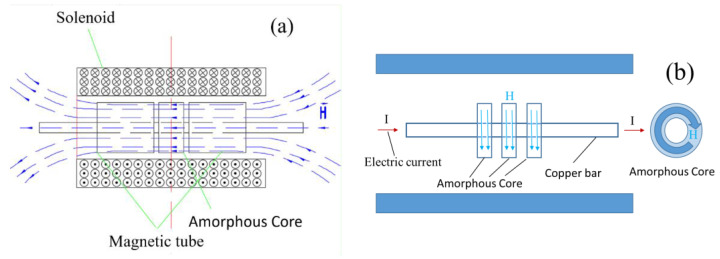
(**a**) Schematic diagram of TFA; (**b**) Schematic diagram of LFA.

**Figure 2 materials-16-05527-f002:**
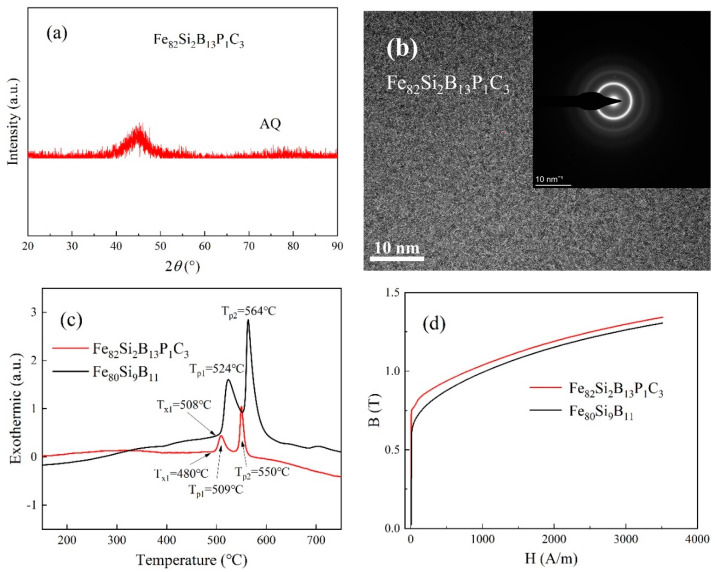
(**a**) XRD curve of as-spun Fe_82_Si_2_B_13_P_1_C_3_; (**b**) HRTEM image and SAED pattern of as-spun Fe_82_Si_2_B_13_P_1_C_3_; (**c**) DSC curves of Fe_80_Si_9_B_11_ and Fe_82_Si_2_B_13_P_1_C_3_; (**d**) Magnetization curves of Fe_80_Si_9_B_11_ and Fe_82_Si_2_B_13_P_1_C_3_.

**Figure 3 materials-16-05527-f003:**
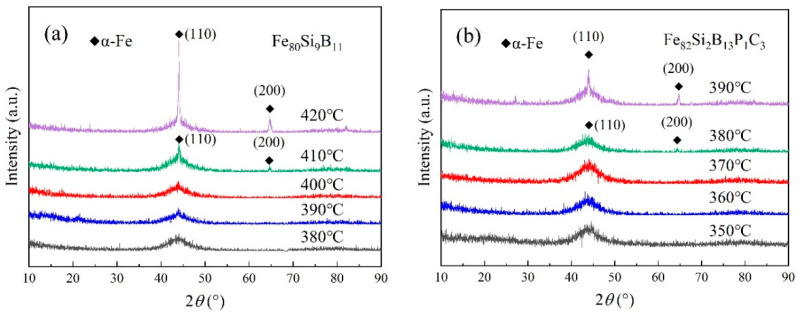
(**a**) XRD curves of Fe_80_Si_9_B_11_ at different annealing temperatures; (**b**) XRD curves of Fe_82_Si_2_B_13_P_1_C_3_ at different annealing temperatures.

**Figure 4 materials-16-05527-f004:**
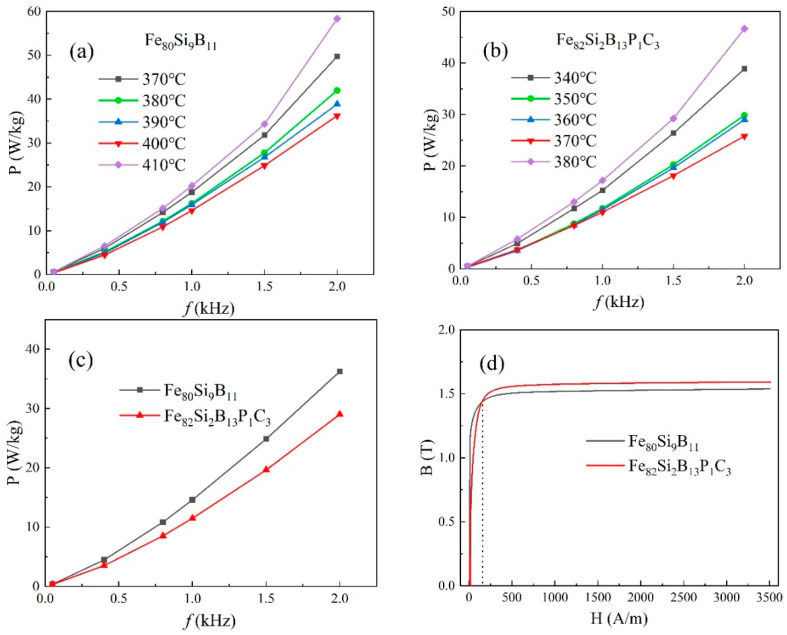
(**a**) Fe_80_Si_9_B_11_ loss curves under different temperatures and frequencies at an excitation of 1.4 T; (**b**) Fe_82_Si_2_B_13_P_1_C_3_ loss curves under different temperatures and frequencies at an excitation of 1.4 T; (**c**) Loss curves under different frequencies for annealed Fe_80_Si_9_B_11_ (400 °C) and Fe_82_Si_2_B_13_P_1_C_3_ (370 °C) at an excitation of 1.4 T; (**d**) Magnetization curves of annealed Fe_80_Si_9_B_11_ (400 °C) and Fe_82_Si_2_B_13_P_1_C_3_ (370 °C).

**Figure 5 materials-16-05527-f005:**
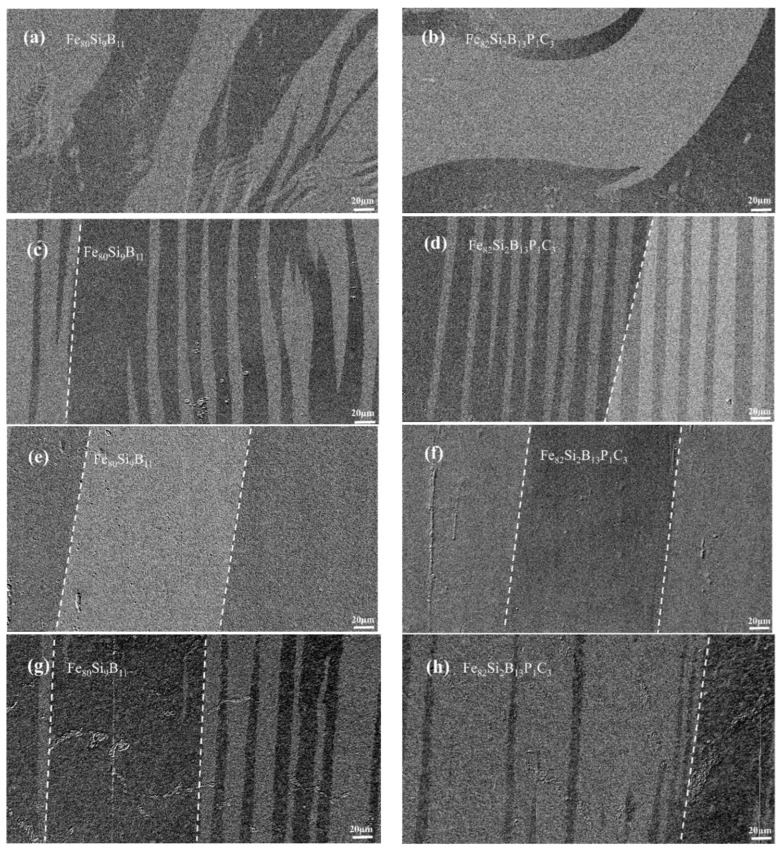
(**a**) Domain morphology of as-spun Fe_80_Si_9_B_11_; (**b**) Domain morphology of as-spun Fe_82_Si_2_B_13_P_1_C_3_, (**c**) Domain morphology of Fe_80_Si_9_B_11_ at 370 °C; (**d**) Domain morphology of Fe_82_Si_2_B_13_P_1_C_3_C3 at 340 °C; (**e**) Domain morphology of Fe_80_Si_9_B_11_ at 400 °C; (**f**) Domain morphology of Fe_82_Si_2_B_13_P_1_C_3_ at 370 °C; (**g**) Domain morphology of Fe_80_Si_9_B_11_ at 410 °C; (**h**) Domain morphology of Fe_82_Si_2_B_13_P_1_C_3_ at 380 °C.

**Figure 6 materials-16-05527-f006:**
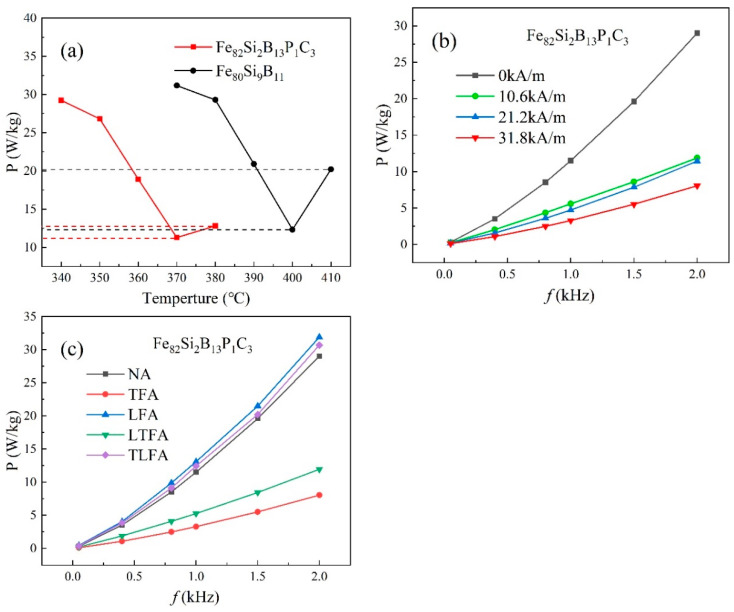
(**a**) TFA losses under different temperatures at 10.6 kA/m, 1.4 T, 2 kHz; (**b**) Losses at 370 °C when the transverse magnetic field was varied; (**c**) Loss curves of NA, TFA, LFA, LTFA and TLFA at 370 °C.

**Figure 7 materials-16-05527-f007:**
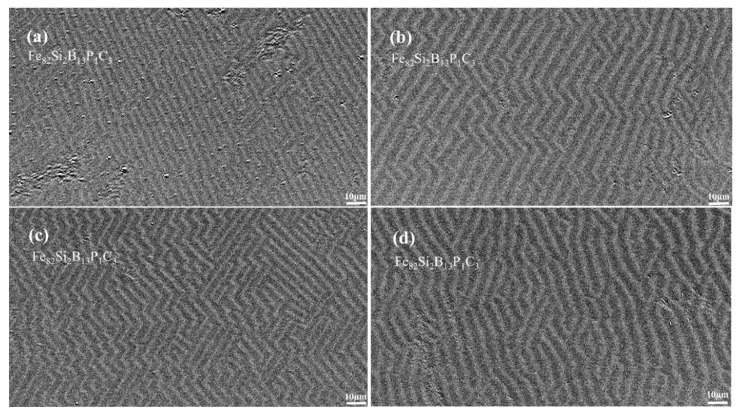
(**a**) TFA domain morphology for Fe_82_Si_2_B_13_P_1_C_3_ annealed at 370 °C; (**b**) LFA domain morphology for Fe_82_Si_2_B_13_P_1_C_3_ annealed at 370 °C; (**c**) LTFA domain morphology for Fe_82_Si_2_B_13_P_1_C_3_ annealed at 370 °C; (**d**) TLFA domain morphology for Fe_82_Si_2_B_13_P_1_C_3_ annealed at 370 °C.

**Figure 8 materials-16-05527-f008:**
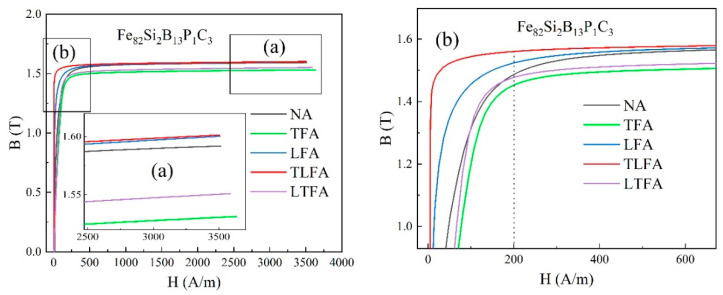
(**a**) Magnetization curves of NA, TFA, LFA, LTFA and TLFA samples annealed at 370 °C; (**b**) Partially enlarged images.

**Table 1 materials-16-05527-t001:** Thermodynamic properties of Fe_82_Si_2_B_13_P_1_C_3_ and Fe_80_Si_9_B_11_ ribbons in as-spun state.

Item	T_x1_ (°C)	T_p1_ (°C)	T_p2_ (°C)	ΔT_p_ = T_p2_ − T_p1_ (°C)
Fe_82_Si_2_B_13_P_1_C_3_	480	509	550	41
Fe_80_Si_9_B_11_	508	524	564	40
Difference (T_Fe82Si2B13P1C3_ − T_Fe80Si9B11_)	−28	−15	−14	1

**Table 2 materials-16-05527-t002:** NA losses and B_m_ values of Fe_82_Si_2_B_13_P_1_C_3_ and Fe_80_Si_9_B_11_ iron cores.

Item	P_1.4T,2kHz_ (W/kg)	P_1.4T,2kHz_ (W/kg)	ΔP_1.4T,2kHz_ (W/kg)	B_3500A/m_ (T)
Fe_82_Si_2_B_13_P_1_C_3_	29 (370 °C)	47 (380 °C)	18	1.59 (380 °C)
Fe_80_Si_9_B_11_	36 (400 °C)	58 (410 °C)	22	1.55 (410 °C)
Difference (Fe_82_Si_2_B_13_P_1_C_3_ − Fe_80_Si_9_B_11_)	−7	−11	−4	0.04

**Table 3 materials-16-05527-t003:** NA losses for Fe_82_Si_2_B_13_P_1_C_3_ and Fe_80_Si_9_B_11_ iron cores.

Item	P_1.4T,2kHz_ (W/kg) 10.6 kA/m	P_1.4T,2kHz_ (W/kg) 10.6 kA/m	P_1.4T,2kHz_ (W/kg) 31.8 kA/m	P_1.4T,2kHz_ (W/kg) LTFA	P_1.4T,2kHz_ (W/kg) TLFA
Fe_82_Si_2_B_13_P_1_C_3_	11.3 (370 °C)	12.8 (380 °C)	8.1 (370 °C)	11.9 (370 °C)	30.7 (370 °C)
Fe_80_Si_9_B_11_	12.3 (400 °C)	20.2 (410 °C)	9.8 (400 °C)	12.5 (400 °C)	32.1 (400 °C)
Difference (P _Fe82Si2B13P1C3_ − P _Fe80Si9B11_)	−1	−7.4	−1.7	−0.6	−1.4

## Data Availability

The data presented in this study are available within the article.

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
