# Peer review of "The Effect of Annealing on the Soft Magnetic Properties and Microstructure of Fe82Si2B13P1C3 Amorphous Iron Cores"

_materials, 2023, doi:10.3390/ma16165527_

Round 1

Reviewer 1 Report

I evaluated the work in great detail. In many respects, it is a very well prepared article, but it has some shortcomings, these parts can be reviewed.

-Introduction part is incomplete as it is, it can be expanded with the literature by detailing. In addition, the desired result with the study should be clearly revealed. For example, why NA-FA annealing was chosen, although there are many methods? If you are not going to mention the others, you can remove other methods from the abstract part.

- The interpretation of the results obtained is sufficient, but there is no clear evaluation about the benefits of annealing applied procedures and the effects of the differences between them in the conclusion part. What exactly is the advantage?

There are some minor errors in the English spelling, it can be revised but generally sufficient.

Author Response

Dear Edit

Based on your question, make the following modifications:

  1. Introduction

Relaxation upon isothermal annealing below crystallization transition tempera-ture is applied to release the residual stress in metallic amorphous [12], to enhance their soft magnetic performance [13]. Ordinarily, the soft magnetic properties of amorphous alloys can be further optimized through appropriate annealing processes such as annealing temperature, heating rate, and holding time [14].The as-quenched amorphous ribbons are usually subjected to magnetic field annealing to improve the soft magnetic properties (SMP) [15]. Magnetic field annealing treatment could bring about a process of atomic-pair ordering and the decline of the coercive field [16]. Mag-netic field annealing had a positive effect on changing the domain sensitive direction and unifying the easy magnetization direction [17]. Magnetic field caused the for-mation of strip domains in the sample for optimal SMP [18]. Magnetic field annealing could promote inner stress release and modulate the domain structure which directly affects the soft-magnetic alloys [19].However, the mechanism of the effect of magnetic field annealing on SMP is not yet clear.

  1. Methods

Sample source: The GFA and soft magnetic properties of as-spun Fe80Si9B(11−x)Px amorphous alloys were improved due to the incorporation of a minor amount of P el-ement. However, the excess of P led to the reduction of iron content and the deteriora-tion of saturation magnetization[20].

Normal annealing (NA): The temperature of the furnace is then raised from room temperature to the designat-ed holding temperature (Fe80Si9B11:370-420°C and Fe82Si2B13P1C3:340-390°C) at a heating rate of 6 °C/min. The temperature is maintained at the holding temperature for a dura-tion of 40 min, followed by furnace cooling until reaching room temperature to allow for sample removal.

Test method: The amorphous nature of the alloy strip was assessed using X-ray diffraction (XRD) with Cu Kα (λ = 0.15406nm) radiation in the range of 30 - 90° taking 0.2 s for one step of 0.02°. FEI Strata 400S focused ion beam scanning electron micros-copy (FIB-SEM) was used to cut the ribbon, and FEI Talos F200s transmission electron microscopy (TEM) was used to observe the cross-section of the ribbon. Thermal pa-rameters including crystallization temperature (Tx) and crystallization peak (Tp) were analyzed by differential thermal analysis (DSC) at a heating rate of 10 ℃/min under the protection of Argon gas and establish an appropriate range of annealing tempera-tures. The loss and hysteresis loop of the amorphous iron core were measured using both direct current (model TD8150) and alternating current (model TK7500) magnetic performance testing instruments. Additionally, a magneto-optical Kerr effect (MOKE, Evico 4-873K/950 MT) microscope was employed to examine the magnetic domain structures.

  1. Conclusion

3)Fe82Si2B13P1C3 amorphous iron cores under an applied excitation of 3500 A/m, the Bm values of the treatment processes were arranged in ascending order as TFA<LTFA<NA<LFA<TLFA. Particularly, the Bm values of LFA and TLFA closely overlapped, both reaching 1.6 T.

  1. grammatical errors

There are revised graphical and grammatical errors,example:Figure 8. (a)Magnetization curves of NA, TFA, LFA, LTFA, and TLFA samples annealed at 370 °C, (b) Partially enlarged images.

Reviewer 2 Report

The research paper titled "Effects of heat treatment processes on soft magnetic properties for Fe82Si2B13P1C3 amorphous iron cores" provides a comprehensive investigation into the impact of normal annealing (NA) and magnetic field annealing (FA) on the soft magnetic properties and microstructure of Fe82Si2B13P1C3 amorphous alloy iron cores. The study employed various methods of magnetic field application, including transverse magnetic field annealing (TFA), longitudinal magnetic field annealing (LFA), transverse magnetic field followed by longitudinal magnetic field annealing (TLFA), and longitudinal magnetic field followed by transverse magnetic field annealing (LTFA). The findings were compared with those of commercially produced Fe80Si9B11.

The paper covers a range of analytical techniques such as differential scanning calorimetry (DSC), transmission electron microscopy (TEM), X-ray diffraction (XRD), magnetic performance testing equipment, and magneto-optical Kerr microscopy, which collectively provide a comprehensive analysis of the annealed samples. The research findings are well-presented and supported by the data obtained from the experimental methods.

One notable result is the reduced loss of P1.4 T, 2 kHz for Fe82Si2B13P1C3, which was achieved by annealing in a transverse magnetic field at 370 °C. This result demonstrated a 17% improvement over Fe80Si9B11, indicating the potential of Fe82Si2B13P1C3 as a soft magnetic material. Additionally, the influence of the longitudinal magnetic field resulted in a more rectangular magnetization curve and an increased coercivity for Fe82Si2B13P1C3, suggesting enhanced magnetic properties compared to Fe80Si9B11.

The paper also provides valuable insights into the microstructural changes occurring during the 370 °C annealing process of the Fe82Si2B13P1C3 amorphous iron core. The gradual dissipation of internal stress and the alignment of impurity domains with the length direction of the strip contributed to the formation of wide strip domains with low resistance and easy magnetization. This phenomenon played a crucial role in reducing the overall loss of the amorphous iron core.

In summary, the research paper contributes to the understanding of the soft magnetic properties and microstructure of Fe82Si2B13P1C3 amorphous iron cores and compares them to commercially produced Fe80Si9B11. The methodology and analysis presented are robust and provide meaningful insights into the effects of different annealing processes. The results have implications for the development of improved soft magnetic materials for various applications.

While the paper presents valuable findings, there are areas that need further attention to improve the overall quality and clarity of the manuscript. Please consider the following revisions:

Clarify the significance and potential applications of Fe82Si2B13P1C3 amorphous iron cores. Highlight how the improved soft magnetic properties make them suitable for specific technological applications, For defining the author can use the study perfomred on various pervious studies some examples are a. https://doi.org/10.1016/j.cja.2022.03.002 , b. https://doi.org/10.2139/ssrn.4188588, c.https://doi.org/10.1016/j.matpr.2021.11.143 , d. DOI 10.1088/2053-1591/ab6acc

Provide more details on the experimental methodology, including sample preparation, annealing parameters, and the specific techniques employed for each analysis. This will help readers understand the experimental setup and reproduce the results if desired.

Include additional discussion on the implications of the results, addressing how the observed changes in soft magnetic properties and microstructure contribute to the overall performance of Fe82Si2B13P1C3 amorphous iron cores.

Provide more context and discussion around the comparison with commercially produced Fe80Si9B11. Highlight the advantages and disadvantages of Fe82Si2B13P1C3 in comparison and discuss potential reasons for the observed differences in soft magnetic properties.

Review the paper for grammatical errors, typos, and inconsistencies in terminology. Ensure clarity and precision throughout the manuscript.

Final Decision: Major Revision Required

 Minor editing of English language required

Author Response

Dear Edit

Based on your question, make the following modifications:

  1. Introduction

Soft magnetic materials and their related devices (inductors, transformers ,and electrical machines) play a key role in the conversion of energy throughout our world. Conversion of electrical power includes the bidirectional flow of energy between sources, storage, and the electrical grid and is accomplished via the use of power electronics. Electrical machines (motors and generators) transform mechanical energy into electrical energy and vice versa .The introduction of wide band gap (WBG) semiconductors is allowing power conversion electronics and motor controllers to operate at much higher frequencies. This reduces the size requirements for passive components (inductors and capacitors) in power electronics and enables more-efficient, high–rotational speed electrical machines [1]. Fe-based amorphous alloys exhibit superior soft magnetic properties including relatively high saturated magnetization, high effective permeability, low coercivity and low core loss especially at middle and high frequency, compared to other traditional soft magnetic materials, like ferrite, silicon steel and permalloy.This kind of alloys with unique structure, excellent magnetic properties and good mechanical properties have attracted much attentions in a variety of emerging science and engineering fields[2].

Relaxation upon isothermal annealing below crystallization transition temperature is applied to release the residual stress in metallic amorphous [12], to enhance their soft magnetic performance [13]. Ordinarily, the soft magnetic properties of amorphous alloys can be further optimized through appropriate annealing processes such as annealing temperature, heating rate, and holding time [14].The as-quenched amorphous ribbons are usually subjected to magnetic field annealing to improve the soft magnetic properties (SMP) [15]. Magnetic field annealing treatment could bring about a process of atomic-pair ordering and the decline of the coercive field [16]. Mag-netic field annealing had a positive effect on changing the domain sensitive direction and unifying the easy magnetization direction [17]. Magnetic field caused the for mation of strip domains in the sample for optimal SMP [18]. Magnetic field annealing could promote inner stress release and modulate the domain structure which directly affects the soft-magnetic alloys [19].However, the mechanism of the effect of magnetic field annealing on SMP is not yet clear.

  1. Methods

Sample source: The GFA and soft magnetic properties of as-spun Fe80Si9B(11−x)Px amorphous alloys were improved due to the incorporation of a minor amount of P element. However, the excess of P led to the reduction of iron content and the deterioration of saturation magnetization [20].

Normal annealing (NA): The temperature of the furnace is then raised from room temperature to the designated holding temperature (Fe80Si9B11:370-420°C and Fe82Si2B13P1C3:340-390°C) at a heating rate of 6 °C/min. The temperature is maintained at the holding temperature for a dura-tion of 40 min, followed by furnace cooling until reaching room temperature to allow for sample removal.

Test method: The amorphous nature of the alloy strip was assessed using X-ray diffraction (XRD) with Cu Kα (λ = 0.15406nm) radiation in the range of 30 - 90° taking 0.2 s for one step of 0.02°. FEI Strata 400S focused ion beam scanning electron micros-copy (FIB-SEM) was used to cut the ribbon, and FEI Talos F200s transmission electron microscopy (TEM) was used to observe the cross-section of the ribbon. Thermal parameters including crystallization temperature (Tx) and crystallization peak (Tp) were analyzed by differential thermal analysis (DSC) at a heating rate of 10 ℃/min under the protection of Argon gas and establish an appropriate range of annealing temperatures. The loss and hysteresis loop of the amorphous iron core were measured using both direct current (model TD8150) and alternating current (model TK7500) magnetic performance testing instruments. Additionally, a magneto-optical Kerr effect (MOKE, Evico 4-873K/950 MT) microscope was employed to examine the magnetic domain structures.

  1. Conclusion

3)Fe82Si2B13P1C3 amorphous iron cores under an applied excitation of 3500 A/m, the Bm values of the treatment processes were arranged in ascending order as TFA<LTFA<NA<LFA<TLFA. Particularly, the Bm values of LFA and TLFA closely overlapped, both reaching 1.6 T.

  1. advantages and disadvantages

In short,Fe82Si2B13P1C3 amorphous iron core demonstrated a lower loss than that of Fe80Si9B11. Fe82Si2B13P1C3 amorphous iron core in the motor has the advantage of high efficiency, high power density and high torque density. Fe82Si2B13P1C3 amorphous iron core exhibited a higher Bm than Fe80Si9B11. Compared with Fe80Si9B11, amorphous iron core possess relatively lower Bm, which hampers the miniature of the devices.

  1. grammatical errors

There are revised graphical and grammatical errors,example:Figure 8. (a)Magnetization curves of NA, TFA, LFA, LTFA, and TLFA samples annealed at 370 °C, (b) Partially enlarged images.

Round 2

Reviewer 2 Report

I carefully evaluated the manuscript's revised version. The authors have significantly modified the updated version in response to the referees' suggestions. The parameters have been clearly defined, and the manuscript has become more fluent. As a result, I feel that the work, in its present form, is of sufficient quality to merit publication in this journal.
